# Effects of Dietary Protein Source and Quantity on Bone Morphology and Body Composition Following a High-Protein Weight-Loss Diet in a Rat Model for Postmenopausal Obesity

**DOI:** 10.3390/nu14112262

**Published:** 2022-05-28

**Authors:** Christian S. Wright, Erica R. Hill, Perla C. Reyes Fernandez, William R. Thompson, Maxime A. Gallant, Wayne W. Campbell, Russell P. Main

**Affiliations:** 1Department of Physical Therapy, School of Health and Human Sciences, Indiana University, Indianapolis, IN 46202, USA; pcfernan@uab.edu (P.C.R.F.); thompwil@iu.edu (W.R.T.); 2Indiana Center for Musculoskeletal Health, Indiana University, Indianapolis, IN 46202, USA; rmain@purdue.edu; 3Department of Nutrition Science, Purdue University, West Lafayette, IN 47907, USA; hill155@purdue.edu (E.R.H.); campbeww@purdue.edu (W.W.C.); 4Department of Physical Therapy, School of Health Professions, The University of Alabama at Birmingham, Birmingham, AL 35294, USA; 5Department of Anatomy and Cell Biology, Indiana University, Indianapolis, IN 46202, USA; 6Department of Basic Medical Sciences, Purdue University, West Lafayette, IN 47907, USA; max@heliohealth.com; 7Weldon School of Biomedical Engineering, Purdue University, West Lafayette, IN 47907, USA

**Keywords:** protein source, high protein, weight loss, bone density, postmenopausal

## Abstract

Higher protein (>30% of total energy, HP)-energy restriction (HP-ER) diets are an effective means to improve body composition and metabolic health. However, weight loss (WL) is associated with bone loss, and the impact of HP-ER diets on bone is mixed and controversial. Recent evidence suggests conflicting outcomes may stem from differences in age, hormonal status, and the predominant source of dietary protein consumed. Therefore, this study investigated the effect of four 12-week energy restriction (ER) diets varying in predominate protein source (beef, milk, soy, casein) and protein quantity (normal protein, NP 15% vs. high, 35%) on bone and body composition outcomes in 32-week-old obese, ovariectomized female rats. Overall, ER decreased body weight, bone quantity (aBMD, aBMC), bone microarchitecture, and body composition parameters. WL was greater with the NP vs. HP-beef and HP-soy diets, and muscle area decreased only with the NP diet. The HP-beef diet exacerbated WL-induced bone loss (increased trabecular separation and endocortical bone formation rates, lower bone retention and trabecular BMC, and more rod-like trabeculae) compared to the HP-soy diet. The HP-milk diet did not augment WL-induced bone loss. Results suggest that specific protein source recommendations may be needed to attenuate the adverse alterations in bone quality following an HP-ER diet in a model of postmenopausal obesity.

## 1. Introduction

With nearly two-thirds of Americans classified as overweight or obese [1], quality of life has dramatically decreased in the wake of increasing metabolic disorders [2]. Energy restriction (ER) is often a recommended strategy for the treatment of obesity [3], and higher protein (HP, >1.2 g·kg^−1^·day^−1^ or >30% of total energy consumed), ER diets are commonly used to lose weight, preserve fat-free mass [4,5,6,7,8], increase thermogenesis [5], and improve lipid lipoprotein profiles [6]. However, body weight is closely related to bone mass [9], and weight loss (WL) decreases bone mineral density (BMD) [10,11,12]. In fact, a 10% reduction in body weight is associated with up to a 2% decrease in whole-body BMD [13,14], which could, in turn, decrease bone strength and increase fracture risk. A recent meta-analysis of randomized controlled trials confirms these findings showing decreased total hip BMD following 6 months of ER in adults (−0.01 to −0.015 g/cm^2^) [15]. Furthermore, while beneficial for WL, controversy exists surrounding the potentially detrimental effects of HP diets on bone.

The proposed mechanism for this theorized protein-induced bone loss has been termed the acid-ash hypothesis [16], where consumption of excess dietary protein increases acid production via oxidation of sulfur-containing amino acids and phosphoproteins, leading to bone demineralization [17,18] and a negative calcium balance [19,20,21]. Though increases in dietary protein during ER are associated with increases in urinary calcium excretion [4,22], dual-stable isotope studies show an increase in intestinal calcium absorption with HP intakes [23,24], mediated through lysine and arginine [25], which can offset these increases in urinary calcium excretion [4,19,26,27,28]. Additionally, increasing total protein intake during ER increases circulating insulin-like growth factor 1 (IGF-1) [29], which stimulates bone formation [30] and has an overall positive impact on BMD [31,32,33,34,35,36]. As a result, HP diets do not impair bone health but potentially offer a mechanical (maintained fat-free mass), biochemical (increased IGF-1), and homeostatic (increased calcium absorption) advantage in preserving bone mass during ER. Yet, even with a recent meta-analysis showing a positive effect of HP diets on BMD during ER [37], confusion persists in the scientific community with studies that show an attenuation [35,36,38], exacerbation [39], or neutral effect [40,41] of HP-ER diets on WL-induced bone loss.

These conflicting outcomes may be partially explained by differences in the predominant source of dietary protein and the use of either an isolated protein compound or a protein-containing food [42,43,44,45,46,47,48,49]. Isolated protein compounds such as whey protein isolate contain a fixed amino acid composition and have little to no additional nutritive or non-nutritive components. Protein-containing foods are whole food sources of dietary protein that possess varying amino acid compositions as well as additional micronutrient, bioactive, and non-nutritive components not found in individual protein compounds. These differences in protein source have the potential to differentially influence bone health [42,45,46,47,48,49,50,51,52,53,54,55,56,57,58], shown by greater increases in serum IGF-1 concentrations with meat versus soy consumption [59] and lower bone turnover markers with a high dairy, HP-ER diet versus a mixed HP-ER diet [32]. More recently, normal-weight middle-aged male rats showed greater increases in cortical volumetric BMD and femoral calcium content with a predominately soy protein concentrate HP-ER diet versus a predominately milk protein concentrate HP-ER diet [60]. Age and hormonal status of the population may also influence the effects of HP-ER diets on bone. Post-menopausal women consuming an HP lean meat-based ER diet had greater decreases in whole-body BMD than post-menopausal women consuming a lacto-vegetarian, normal protein (NP, 0.8 g·kg^−1^·day^−1^) ER diet [39]. These effects, however, were not found in middle-aged men and women of the same cohort [61], suggesting these detrimental effects may be influenced by age and hormonal status.

Currently, there is a lack of consistency in the literature concerning the effects of a HP-ER diet on bone and the potential influence of protein sources. Therefore, our objective was to investigate the effects of protein source (beef, milk, and soy) and protein quantity (NP, 15% energy vs. HP, 35% energy) on bone and body composition outcomes following a 12-week ER diet in obese ovariectomized female Sprague Dawley rats. We hypothesized that (1) The HP predominately milk-based diet, which is rich in calcium and other potential bioactive bone-sparing compounds, would show the greatest attenuation of WL-induced bone loss and (2) The HP predominately lean-beef diet would show the greatest exacerbation of WL-induced bone loss in comparison to other energy-restricted diets.

## 2. Materials and Methods

### 2.1. Animals

Twenty-week old Sprague Dawley female rats (*n* = 40) were purchased from Charles River Laboratories (Wilmington, MA, USA) and housed individually in standard polycarbonate cages under temperature-controlled environmental conditions (12 h light:dark period). Upon arrival, rats were given free access to purified water and the standardized AIN-93M rodent diet [62] (Research Diets Inc., New Brunswick, NJ, USA, product# D10012M). Following the one-week acclimation period, rats underwent ovariectomy (OVX, *n* = 36) or sham operation (Sham, *n* = 4) before consuming a high fat (45% of energy), high sugar (22% of energy) ad libitum diet (HFHS, Supplemental Appendix A, Research Diets Inc., product# D15053103) for 12 weeks to induce obesity [63]. Following the 12-week HFHS, OVX (*n* = 4) and sham-operated (*n* = 4) rodents were sacrificed to confirm ovariectomy and establish a baseline group for subsequent WL analyses. The remaining 32-week-old OVX obese rodents (*n* = 32) were then randomized (Microsoft Corporation 2010, Microsoft Excel; Redmond, WA USA; randomization function) into one of four ER diets (40% energy reduction, *n* = 8) with varying protein concentrations/source (WL intervention). The study’s experimental timeline and time of data acquisition are described in Figure 1. All experimental procedures were approved by Purdue University’s Animal Care and Use Committee (protocol# 1412001176, approved 13 January 2017).

### 2.2. Weight Loss Dietary Intervention

Following the 12-week HFHS diet and baseline sacrifice (*n* = 8), rodents (*n* = 32) were randomized to 1 of 4 energy-restriction (ER) diets for 12 weeks to induce weight loss. The ER diets varied in a predominant source of dietary protein or protein quantity and were divided into the following diets (*n* = 8 per diet): (1) HP-ER (35% energy from protein) freeze-dried lean beef diet (HP-beef), (2) HP-ER milk protein isolate diet (HP-milk), (3) HP-ER soy protein isolate diet (HP-soy), and (4) Normal protein (NP, 15% energy from protein) ER diet (NP-Control). Throughout the 12-week intervention, the individually housed rodents consumed a daily allotment of food that accounted for 60% of their average daily energy intake during the HFHS diet, resulting in an estimated 40% ER from baseline. Food intake was monitored daily throughout the 12-week intervention to maintain 40% ER from baseline.

### 2.3. Diet Formulation

ER diets were formulated and manufactured by Research Diets, Inc. (New Brunswick, NJ, USA) from the standardized AIN-93M rodent diet [62]. While the AIN-93M rodent diet served as the NP-control diet (15% energy from protein, 76% energy from carbohydrates), HP-ER diets were manufactured by Research Diets to replace 20% of the energy from corn starch (56% energy from carbohydrates) with either freeze-dried lean beef (HP-beef), milk protein isolate (HP-milk), or soy protein isolate (HP-soy) to equal an additional 20% of dietary protein to each diet (35% energy from protein). Therefore, all HP-ER diets contained 15% of their dietary protein from the standardized AIN-93M diet (casein) plus an additional 20% of dietary protein from their respective protein source, creating a mixed but predominantly beef-, milk-, or soy-based diet. This approach was akin to two previous WL interventions conducted in postmenopausal women by Campbell and colleagues, wherein the substitution of carbohydrates for striated tissue-based protein sources (+0.6 g·kg^−1^·day^−1^ from beef, pork, or chicken) lowered total body BMD versus a mixed NP-ER diet (0.8 g·kg^−1^·day^−1^) [39]. Clinically, a diet with 35% of total energy intake from dietary protein is considered the upper limit of the acceptable macronutrient distribution range (AMDR; 10–35% dietary protein) [64], which is often surpassed during times of ER [65,66].

To incorporate a whole food source of dietary protein into an ER diet, a novel freeze-dried lean beef powder was created in-house. Lean ground beef (94% fat-free pectoral muscle) was purchased locally (West Lafayette, IN, USA) and slowly dehydrated over five days by freeze-drying methods (−20 °C to 10 °C at 120 mTorr; Virtis GPFD 24DX48, Stone Ridge, NY, USA). Freeze-dried lean beef was then ground into a fine powder utilizing a Food Service-Grade pulverizer and sent for nutritional analyses (University of Missouri Agricultural Experiment Station Chemical Laboratories, Columbia, MO, USA) before the beef powder was sent to Research Diets for incorporation. Macronutrient and amino acid analysis of freeze-dried lean beef powder is presented in Supplemental Appendix A. IdaPro^®^ milk protein isolate (85% Protein DMB; derived from Grade A milk, reduced lactose, Idaho Milk Products, Jerome, ID, USA) and SUPRO^®^ 661 soy protein isolate (The Solae Company, Geneva, Switzerland) were purchased individually and incorporated to meet diet specifications. To prevent micronutrient deficiencies during energy restriction, mineral and vitamin concentrations were adjusted to account for a 40% energy restriction. Diet formulation and compositions are found in Table 1 and Table 2, respectively.

#### 2.3.1. Tissue Collection

Rats were euthanized via deep anesthetization and exsanguination by cardiac puncture. A secondary method of euthanasia, cervical dislocation, was employed following exsanguination to ensure cardiac arrest and loss of brain function. The right femur and fifth lumbar vertebrae were cleaned of soft tissue and fixed in 10% neutral buffered formalin for 24 h at 4 °C before storing in 70% ethanol. The left femur was cleaned of soft tissue and frozen at −20 °C until mechanical testing.

#### 2.3.2. Body Composition

Two cross-sectional images of the L4-5 intervertebral disc were taken at baseline and post-intervention by peripheral quantitative computed tomography (pQCT, XCT-2000; Stratec Medizintechnic, Pforzheim, Germany) to assess changes in fat (mm^2^), muscle (mm^2^), muscle density (MD, g/cm^2^), and intramuscular adipose tissue (IMAT, mm^2^) over the 12-week WL intervention. Rats were anesthetized with isoflurane (2.5%, IsoFlo, Abbott Laboratories, Chicago, IL, USA) mixed with oxygen (O_2_, 1.5 L/min) for ~5 min, including induction and scanning time [67], and immobilized dorsally in a holding chamber with the head facing away from pQCT gantry. Once properly aligned, scout scans were conducted to identify the location of the fourth and fifth lumbar vertebrae before two cross-sectional images (spaced 2.2 mm apart, 0.121 voxel size) were acquired at the L4-5 intervertebral disc. Baseline and post-intervention muscle, fat, MD, and IMAT values were generated following the average of two tomographic slices generated from multiple analyses of the total abdominal and muscle areas. To quantify muscle, both fat and bone (B) were individually quantified and subtracted from the total abdominal area. Fat was then quantified by contour mode 4 and peel mode 2 using a threshold of 50 mg/cm^3^, while B was quantified using a threshold of 300 mg/cm^3^. To assess MD, muscle bone (MB) and density (MBD) were first quantified and separated from fat using a threshold of 50 mg/cm^3^. Additional indices of B and bone density (BD) were further quantified using a threshold of 300 mg/cm^3^, allowing MD to be calculated from the following equation: Muscle Density = [(MBD + MB/B) − BD]/[(MB − B)/B]. To assess IMAT, the adipose tissue located beneath the fascia and in-between or inside muscle fibers, a contour mode 3 and peel mode 2 non-filtered analysis using a threshold of −150 mg/cm^3^ and an inner threshold of 40 mg/cm^3^ was used to separate total fat from muscle, and B. Fat (threshold −101 mg/cm^3^, inner threshold 40 mg/cm^3^) and bone marrow (threshold 710 mg/cm^3^, inner threshold −101 mg/cm^3^) were then subtracted from total fat to produce an IMAT value. Similar in vivo analyses of the abdominal mid-section of rodents have previously been conducted to assess total adiposity and have shown high correlations to whole-body scans and ex vivo tissue weighing [68]. Quality assurance was verified with daily phantom scans. Body weight was measured twice a week to track the progress of WL.

#### 2.3.3. Dual-Energy X-ray Absorptiometry (DXA)

Longitudinal DXA measurements were taken prior to ovariectomy, at baseline and post-intervention, on live rodents using the Lunar PIXImus densitometer (Lunar software ver. 2.00, Fitchburg, WI, USA) to assess changes in bone quantity. Rats were anesthetized (2.5% isoflurane, 1.5 L/min O_2_) and placed in a prone position on the scanner to collect measurements of areal bone mineral content (aBMC) and areal bone mineral density (aBMD) of the right femur and lumbar spine (L1–L4).

#### 2.3.4. Micro-Computed Tomography (µCT)

Bone microarchitecture of the right femur and fifth lumbar vertebrae were determined by ex vivo µCT (μCT40 Scanner; Scanco Medical, Switzerland) following a similar protocol as in previous studies [69,70,71]. Before the initiation of all scanning procedures, the total lengths of the right femora and fifth lumbar vertebrae were measured by μCT to determine the region of interest and standardize the scanning protocol between samples. Distal metaphysis and midshaft diaphysis of the right femora were scanned using a voxel size of 16 μm^3^ (X-ray tube potential (peak): 70 kVp, 114 μAs; no frame averaging) to quantify cancellous (integration time: 300 ms) and cortical bone (integration time: 190 ms), respectively. Scanning of the distal metaphysis began at 11% of bone length, proximal to the distal end, and proceeded proximally to the 21% bone site (10% of total bone length, 240–260 slices). The diaphyseal volume of interest was centered at the midshaft and extended for 25 slices both proximally and distally (50 slices, ~2% of total bone length). Fifth lumbar vertebrae were scanned using a voxel size of 20 μm^3^ (70 kVP, 114 μAs; integration time: 300 ms; no frame averaging) beginning at the 50% bone site and extending both proximally and distally for 33% of the total bone length (66% of vertebral body). Single cortical (6700) and cancellous (4400) thresholds were defined similarly to previous studies [71].

Parameters assessed for cancellous bone included volumetric cancellous BMD (VcaBMD), trabecular BMC (Tb BMC), trabecular thickness (Tb.Th), trabecular separation (Tb.Sp), trabecular bone volume fraction (BV/TV), trabecular number (Tb.N), structural model index (SMI), and connectivity density (Conn.D). Midshaft cortical bone analyses included volumetric cortical BMD (VctBMD), total area (Tt.Ar), cortical bone area (Ct.Ar), cortical area fraction (Ct.Ar/Tt.Ar), marrow area (Ma.Ar), cortical thickness (Ct.Th), and polar moment of inertia (pMOI). Data acquisition, analysis, and nomenclature followed JBMR guidelines [72].

#### 2.3.5. Mechanical Testing

Parameters related to whole-bone strength were measured using three-point bending tests on isolated femora as previously described [73]. Briefly, each femur was loaded at the midshaft to failure in a monotonic fashion, during which force and displacement measures were collected every 0.01 s. Ultimate force, energy to ultimate failure, and stiffness were calculated from the force/displacement curves using standard equations [74].

#### 2.3.6. Histological Processing

Femora were dehydrated in graded alcohols, cleared in xylene, and embedded in methylmethacrylate following standard protocols [75]. For static histomorphometry measurements, thick-cut sagittal sections were taken (Leica RM 2255 semi-automated rotary microtome) from the distal end of the femora and manually ground down to ~30 μm. MMA-embedded, thin sections were then deplasticized in acetone and stained by two different procedures: (1) a modification of the von Kossa/MacNeal’s (VKM) Tetrachrome protocol [76] and (2) a tartrate-acid resistant acid phosphatase (TRAP) stain [77]. For VKM staining for osteoblast outcomes, mineralized bone was stained using the Von Kossa silver method and the unmineralized tissue was counterstained with MacNeal’s tetrachrome. Due to sample integrity, only measurements of tissue area (TA), bone surface (BS), osteoid width (OW), and osteoid surface (OS) can be reliability reported. For TRAP staining for osteoclasts, sections were pre-incubated in acetate buffer (0.2 M, pH = 5.0), rinsed, and incubated in a warmed acid phosphatase solution. Sections then were counterstained with Gill’s Hematoxylin No. 3, air dried, and cover-slipped with an aqueous-based mounting media. Histomorphometric analyses were performed (OsteoMetrics Inc., Decatur, GA, USA), and standard nomenclature was applied (BS, bone surface; E.Pm, eroded perimeter; ES, eroded surface; N.Oc, osteoclast number; Oc.S, osteoclast surface; O.S, osteoid surface; O.Th, osteoid thickness) [78].

For dynamic histomorphometry analyses, rats were injected subcutaneously with calcein (20 mg/kg) and alizarin red (20 mg/kg) at 13 and 3 days prior to sacrifice, respectively. In order to measure bone formation, thick-cut cross-sectional slices were taken from the femoral midshaft, manually ground, and digitally imaged on a fluorescent microscope. Digital images were imported into ImagePro Express (Media Cybernetics, Inc., Gaithersburg, MD, USA), and the following histomorphometric measurements were recorded for the periosteal and endocortical surface: total perimeter, single-label perimeter (sL.Pm), double-label area (dL.Ar) and perimeter (dL.Pm), total bone and marrow area. The following results were calculated: mineral apposition rate (MAR = dL.Ar/dL.Pm/9 day), mineralizing surface [MS/BS = (0.5 × sL.Pm + dL.Pm)/total perimeter × 100], and bone formation rate (BFR/BS = MAR × MS/BS × 3.65) [78]. All measurements were collected such that the operator was blinded to treatment. Histology sections, staining, and analyses were conducted by the Histology Core Facilities within the Indiana Center for Musculoskeletal Health.

#### 2.3.7. Statistical Analysis

No differences in body weight, body composition, or skeletal outcomes were observed between OVX rodents at baseline, confirming a lack of confounding variables prior to WL intervention. Given the close relationship between body weight and bone, baseline body weight was used as a covariate for all bone quantity and quality analyses. Outliers were detected in measurements of bone quantity and bone quality and were excluded according to the outlier labeling rule [79]. Repeated Measures ANOVA was performed to determine the main effects of WL and diet (HP-beef, HP-milk, HP-soy, and Control) over time on body weight, body composition, and bone quantity measurements (DEXA outcomes). Two-way ANOVA were performed to determine the main effect of WL (Baseline vs. ER groups) and diet (NP-Control vs. HP-beef vs. HP-milk vs. HP-soy) on bone quality and bone turnover outcomes. One-way ANOVA was performed to determine differences in femoral strength. Post-hoc analyses of significant interactions between baseline and ER diets or among ER diets were carried out by Tukey’s HSD. Statistical analyses were performed using GraphPad Prism (version 9.3.1 for Windows, San Diego, CA, USA). Data are presented as mean ± SEM unless otherwise specified, and a *p* value of <0.05 is considered statistically significant.

## 3. Results

### 3.1. Body Weight and Composition

ER decreased body weight in all groups (*p* < 0.001), losing on average 109 ± 19 g or 26 ± 4% of baseline body weight over the 12-week intervention (Figure 2). WL altered body composition by decreasing fat area (*p* < 0.001) and IMAT (*p* < 0.001), which subsequently increased muscle density (*p* < 0.0001) in all groups. Protein source and protein quantity differentially influenced changes in body composition over the 12-week intervention. The NP-control diet showed greater decreases in body weight (−124 ± 6 g) versus the HP-beef diet (−103 ± 6 g, *p* = 0.022) or HP-soy diet (−98 ± 5 g, *p* = 0.02), due largely to a significant decrease in muscle area following WL (−34 ± 11 mm^2^, *p* = 0.048). All HP-ER diets showed a lack of change in muscle area over the 12-week intervention. Though WL decreased fat area in all groups, the HP-milk diet showed greater decreases in fat area in comparison to the HP-soy diet (*p* = 0.012) (Table 3).

### 3.2. Areal BMD and BMC of Femur and Lumbar Spine

Changes in areal BMD and BMC of the right femur and lumbar spine (L1–L4) can be found in Figure 3 and Table 4. WL decreased lumbar spine aBMD and aBMC (*p* < 0.001) as well as femur aBMD (*p* < 0.001) over the 12-week intervention. Neither protein source nor protein quantity influenced changes in bone mass.

### 3.3. Microarchitecture of Femur and 5th Lumbar Vertebrae

Changes in bone microarchitecture and the main effect of WL relative to baseline can be found in Figure 4 and Supplemental Appendix A. WL decreased cancellous bone at the distal femoral metaphysis, showing decreased trabecular BMC, BV/TV, trabecular number, and connectivity density, as well as a higher SMI (more rod-like) (*p* < 0.05), which suggest a decrease in mechanical competence. WL did not influence measurements at the midshaft femur or the fifth lumbar vertebra.

Relative to baseline, protein source and protein quantity differentially influenced changes in bone microarchitecture following WL intervention (Figure 4, Supplemental Appendix A). The HP-beef diet increased trabecular separation at the distal femoral metaphysis (*p* = 0.009), while the HP-soy diet prevented a change in SMI at the distal femoral metaphysis but decreased cortical area fraction (*p* = 0.036) and increased marrow area (*p* = 0.033) at the femoral midshaft versus baseline. For the vertebral body, the HP-beef diet showed a trend for lower trabecular BMD (*p* = 0.068) and BV/TV (*p* = 0.059) in comparison to baseline. No other changes in microarchitecture were observed over the 12-week intervention relative to baseline.

Following 12 weeks of ER, comparisons between HP diets revealed differential effects of protein source on microarchitecture measurements at the distal metaphysis (Figure 4 and Table 5). The HP-soy diet retained more BV/TV (*p* = 0.028) and trabecular BMC (*p* = 0.009), and lowered SMI (more plate-like, *p* = 0.025) in comparison to the HP-beef diet. A trend for greater volumetric cortical BMD at the midshaft was also observed in the HP-milk diet versus the HP-soy diet (*p* = 0.074) (Table 5). Protein quantity did not influence bone microarchitecture post-intervention.

### 3.4. Femoral Strength and Histomorphometry

Mechanical properties of femora following 12 weeks of ER were tested utilizing ex vivo 3-point bending, and the results are shown in Figure 5. Neither protein source nor protein quantity differentially influenced femoral strength.

The main effects of WL and diet on dynamic and static histomorphometry measurements are shown in Table 6. Relative to baseline, endocortical BFR/BS increased following WL (main effect of WL, *p* = 0.016), with subsequent post-hoc analyses revealing only the HP-beef diet increased endocortical BFR/BS following ER (*p* = 0.032). Endocortical MS/BS appeared to increase with WL (main effect of WL, *p* = 0.019) and be influenced by diet (*p* = 0.047); however, subsequent post-hoc analyses revealed only trends for these effects. The NP diet and HP-beef diet showed trends for greater endocortical MS/BS versus baseline (NP diet, *p* = 0.073; HP-beef diet, *p* = 0.068) and the HP-soy diet showed trends for lower endocortical MS/BS versus NP diet (*p* = 0.099) and HP-beef diet (*p* = 0.093). Neither WL nor diet influenced other measurements of bone formation.

WL influenced histological measurements of osteoclast activity but not osteoblast activity (Table 6). Relative to baseline, WL decreased BS (main effect of WL, *p* = 0.018), driven by a decrease in BS with the NP-control diet (*p* = 0.016) and the HP-beef diet (*p* = 0.037) per post-hoc analyses. In a similar fashion, eroded perimeter increased with WL (main effect of WL, *p* = 0.041) due to the NP-control diet (*p* = 0.049) and the HP-beef diet (*p* = 0.033) in subsequent post-hoc analyses. These indicators of increased bone resorption, however, failed to remain significant after adjusting for total BS (ES/BS, *p* = 0.509). Osteoclast number was also increased following WL relative to baseline (main effect of WL, *p* = 0.017), driven by an increase in osteoclast number with the HP-beef diet (*p* = 0.013) and the HP-milk diet (*p* = 0.017) per post-hoc analyses. However, this increased osteoclast number did not result in increased osteoclast activity relative to BS (Oc.S/BS, *p* = 0.213) nor eroded perimeter (N.Oc/E.Pm, *p* = 0.405).

## 4. Discussion

The current study confirms the negative effects of WL on bone morphology and highlights the differential effects of protein source and protein quantity on bone morphology and body composition during ER. Overall, WL reduced bone quantity and impaired bone microarchitecture and metabolism, regardless of group allocation. These results support previous clinical [15] and animal [80] findings, showcasing a detrimental effect of WL on bone-related outcomes. Though bone quantity decreased universally with WL, WL-induced impairments in bone microarchitecture were largely isolated to the distal femoral metaphysis. This differential response to WL based upon skeletal region could be the result of (1) differences in bone turnover rates between cancellous and cortical bone and/or (2) the impact of weight loss on femoral versus vertebral mechanical unloading. Given that bone turnover rates are higher for cancellous versus cortical bone [81], any changes in bone microarchitecture are more likely to occur in cancellous bone. Rats are quadrupeds and distribute their body weight evenly among four limbs while the spinal cord runs perpendicularly to gravitational forces. Due to these differences in habitual gravitational forces, WL could result in a greater degree of mechanical unloading for femora versus vertebrae. Results from the current study highlight these differences, with WL-induced bone loss primarily occurring at the distal metaphysis versus the midshaft diaphysis or vertebral body. WL-induced decreases in bone quantity were noted for the lumbar spine (aBMD and aBMC, *p* < 0.0001); however, differences in both the number and location of the vertebrae (L1–L4 versus L5) could explain these results. Despite differences in skeletal region, WL led to decreased bone morphology in ovariectomized Sprague Dawley females following 12 weeks of ER.

In agreement with our hypothesis and a previously conducted clinical study, the HP-beef diet exacerbated WL-induced bone loss, showing increased trabecular separation and higher endocortical bone turnover rates versus baseline. This suggests an intrinsic property of lean beef, which negatively influences postmenopausal bone health during ER. Campbell and colleagues were the first to highlight this detrimental effect of increased lean meat consumption during ER on postmenopausal bone mass, as women randomized to a HP, lean meat-based diet showed greater decreases in total body BMD than those consuming a NP, lacto-vegetarian diet following WL [39]. Such results conflict with Sukumar et al. (2011) [29], where a HP WL diet showed bone-sparing effects in comparison to a NP WL diet in postmenopausal women. The HP WL diet not only attenuated WL-induced decreases in bone quantity and microarchitecture but also decreased bone resorption markers and increased IGF-1 concentrations [29]. Such contrasting results occurred despite similarities between the two studies, including age (58 years vs. 60 years), population (postmenopausal women), degree of ER (−600 kcal vs. −750 kcal/d), calcium intakes (1.2 g/d vs. 1 g/d), and total protein intakes (HP: 1.4 g/kg BW/d vs. NP: 0.8 g/kg BW/d). However, unlike Campbell et al. (2010) [39], Sukumar et al. (2011) utilized a mixed HP WL diet where multiple sources of dietary protein were consumed to meet total protein intake requirements [29]. As such, the detrimental effect of a HP lean meat-based WL diet identified by Campbell et al. (2010) could largely be attributable to the predominant source of dietary protein [39]. The current animal study supports this assumption as the HP-beef diet increased trabecular separation and increased endocortical bone turnover rates following WL in comparison to baseline. The exact mechanism behind this meat-induced alteration in bone quality and metabolism following WL in postmenopausal women is currently unknown. However, future research endeavors should strive to identify the diet’s underlining nutritional/bioactive components instigating these responses and the role of both age and hormonal status in this process.

In addition to exacerbating WL-induced bone loss, post-intervention values revealed a lower trabecular BMC, trabecular bone volume fraction, and more rodlike trabeculae with the HP-beef diet in comparison to the HP-soy diet. Though the exact mechanism behind these detrimental effects of the HP-beef diet is still unclear, the lack of change in structural modeling index following WL in comparison to baseline and the attenuated decrease in bone quality with the HP-soy diet is consistent with the effects of isoflavones on bone turnover markers and postmenopausal bone loss [82]. Isoflavones are naturally occurring isoflavonoids, a phenolic compound of the flavonoid family, which possess a similar ring structure to estrogen that allows it to bind to estrogen receptors and elicit estrogen-like regulation of bone turnover [83]. Daidzein and genistein make up the majority of isoflavones found in soy protein and are shown to decrease bone resorption and increase bone formation [84] by stimulating osteoblast differentiation and inhibiting osteoclast formation [85]. The HP-soy diet did not contain purified genistein or daidzein but did provide 371 g of total isoflavone per kilogram of diet or 3.3 to 5.2 mg of total isoflavones per day. Though energy restriction resulted in bone loss globally, the HP-soy diet showed an attenuation of WL-induced bone loss and a lack of change in osteoblast and osteoclast activity, which is similar to isoflavone’s mediation of ovariectomy-induced bone loss [82]. Despite these beneficial effects on cancellous bone, the HP-soy diet also decreased relative cortical bone area and increased marrow area following WL. Differences in bone turnover, bone metabolism, and both mechanical and material properties have long been reported between cancellous and cortical bone [86,87,88,89,90]. The observed lack of change in cancellous bone but impairments in cortical bone following WL with the HP-soy diet could be attributed to the effects of isoflavones on estrogen receptor activity. Mice lacking the estrogen receptor-alpha in macrophages and osteoclasts showed decreased cortical bone but no change in cancellous bone following ovariectomy [91]. These results suggest that estrogen and estrogen-like compounds such as isoflavones can differentially influence bone resorption depending upon (1) bone type, (2) skeletal region, and (3) estrogen receptor concentrations. However, whether the HP-soy diet itself or its increased isoflavone content resulted in these differential effects between cancellous and cortical bone following WL was unclear and may be dependent upon age and hormonal status.

Contrary to our hypothesis, the HP-milk diet did not alter WL-induced decreases in bone quantity, microarchitecture, or metabolism. Though the HP-milk diet does show a trend for greater volumetric cortical BMD at the midshaft post-intervention in comparison to the HP-soy diet, whether this potential difference is a result of the HP-milk or the HP-soy diet is unclear. In contrast, a 16-week HP, milk-based WL diet attenuated decreases in total body, lumbar spine, and total hip BMD in comparison to a NP WL diet in overweight middle-aged adults [33]. Similarly, and independent of total protein intake, a HP milk-based WL diet showed lower bone turnover markers in comparison to an HP mixed WL diet, further suggesting additional benefits of dairy consumption for the preservation of bone during WL outside of dietary protein [35]. However, despite the clinical evidence, the HP-milk diet did not augment WL-induced bone loss. The HP-milk diet did, however, result in greater decreases in fat area in comparison to other HP WL diets (Table 3). Such results are consistent with previous findings showing a greater loss of fat mass during WL and “anti-obesity” effects during weight maintenance with increased dairy consumption [92,93]. These “anti-obesity” effects are believed to be the result of increased calcium intake, which decreases circulating 1,25-dihydroxyvitamin D [1,25-(OH_2_)D_3_] concentrations and may, in turn, stimulate lipolysis and inhibit lipogenesis in adipocytes [94]. Increasing calcium intake via dairy consumption is also shown to increase total fecal fat excretion and aid in WL [95]. Therefore, although the HP-milk diet did not influence WL-induced decreases in bone quantity, microarchitecture, or metabolism, increasing protein intake from dairy sources during WL may result in greater improvements in body composition than other sources of dietary protein.

In contrast to previous clinical findings [29,31,32,33,34,35,36,38] and contrary to our hypothesis, higher protein intakes (35% energy from protein) did not attenuate WL-induced bone loss in comparison to normal protein controls (15% energy from protein). All four ER diets decreased measurements of bone quantity and microarchitecture over the 12-week intervention, as the HP diets offered no advantage to the preservation of bone mass during ER in comparison to the NP-Control. The HP-ER diets did, however, statistically maintain muscle mass throughout the 12-week intervention, while significant decreases over time were observed with the NP-Control diet. This retention of lean muscle mass theoretically provided a mechanical advantage to the bone during ER; however, this was not observed. While protein quantity did not influence changes in bone quantity, microarchitecture, or bone metabolism following WL, differential effects were still observed between the HP-ER diets, suggesting protein source influences the effects of a HP diet on bone during ER. Though most of the clinical data suggest a beneficial effect of HP diets on bone during ER [37], results from the current study suggest that protein source may play an equally important role in the effects of dietary protein on bone during ER.

The strengths of the current study include the inherent objectivity and highly controlled nature of animal research, plus the use of various imaging and histological modalities to identify changes in bone. There are admittedly some limitations. Given the novelty of the research question, power calculations for the derivation of both total and group sample sizes could not be carried out *a priori*. Therefore, the current analysis may be underpowered to detect all other possible effects of protein source and protein quantity on bone. Additionally, resources were unavailable to properly measure circulating bone turnover markers, which could have provided greater insights into bone cell activity and complemented the study’s histological measurements.

## 5. Conclusions

In conclusion, the current animal study further supports the detrimental effect of WL on bone morphology and suggests a differential effect of protein source and protein quantity on bone morphology and body composition in a rat model of postmenopausal obesity. Results reiterate previous clinical observations in postmenopausal women showing an exacerbation of WL-induced bone loss with a HP, beef-based WL diet with significant changes in bone microarchitecture and turnover. Subsequent analytical and histological analyses are needed to identify the inherent nutritive or bioactive compounds associated with each protein source and their subsequent actions on bone metabolism. Though HP WL diets can elicit many advantageous WL and health-related benefits, specific protein source recommendations may be needed to optimize the retention of bone health during WL, particularly in postmenopausal obese women.

## Figures and Tables

**Figure 1 nutrients-14-02262-f001:**
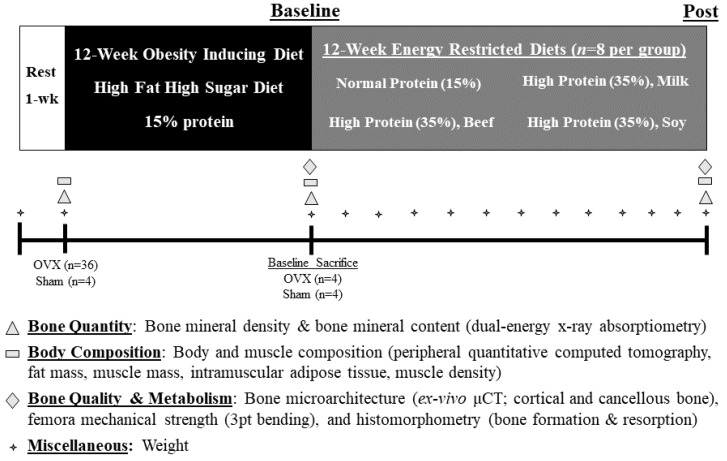
Experimental Design.

**Figure 2 nutrients-14-02262-f002:**
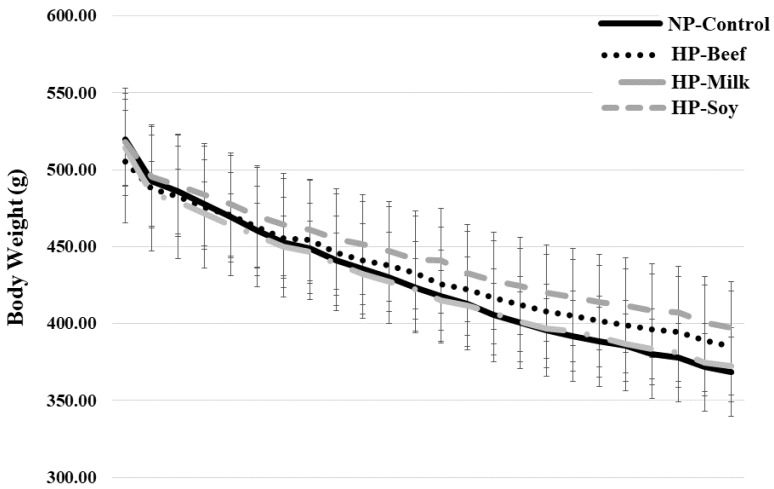
Changes in body weight over the 12-week weight loss intervention in a rat model of postmenopausal obesity. Energy restriction decreased body weight over the 12-week intervention, losing on average 109 ± 19 g or 26 ± 4% of baseline body weight.

**Figure 3 nutrients-14-02262-f003:**
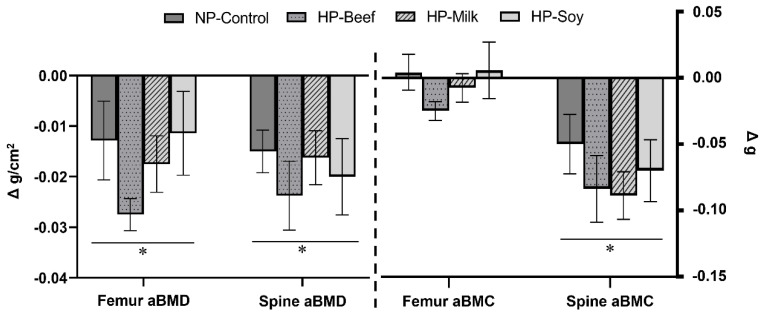
Changes in bone quantity (aBMD/BMC) of total right femur and lumbar spine over the 12-week weight loss intervention in a rat model of postmenopausal obesity. Weight loss decreased measurements of bone quantity independent of protein source and protein quantity. Mean delta values (Post-Baseline) ± SEM; Repeated Measures ANOVA, * Denotes significant effect of weight loss, *p* < 0.05.

**Figure 4 nutrients-14-02262-f004:**
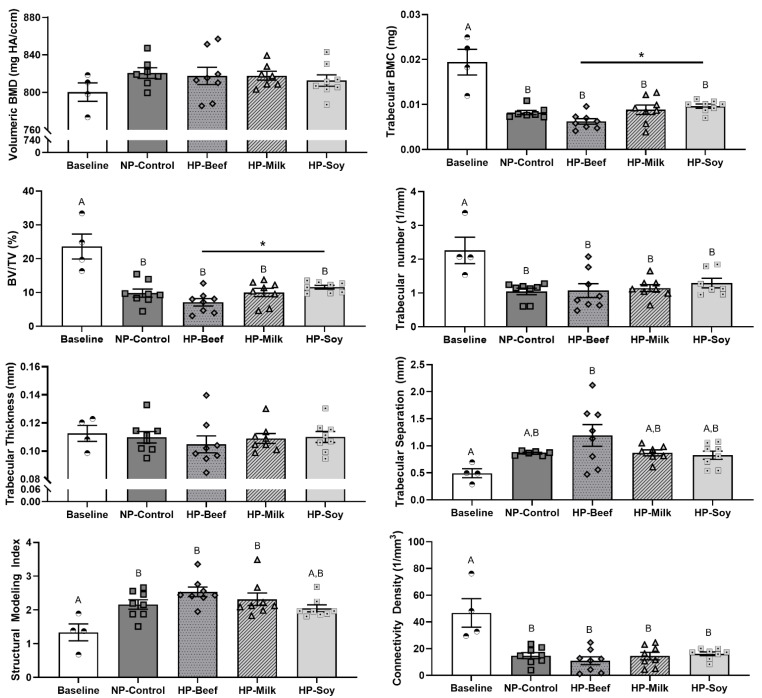
Micro-computed tomography measurements of distal femoral metaphysis: Main effect of weight loss and diet. Weight loss decreased measurements of bone microarchitecture at the distal femoral metaphysis, including trabecular BMC, BV/TV, trabecular number, and connectivity density, while also increasing structural modeling index. Within energy-restricted diets, the HP-soy diet retained more BV/TV and trabecular BMC than the HP-beef diet. Mean ± SEM; BV/TV, trabecular bone volume fraction. Two-way ANOVA were performed; lettering denotes significant main effect of weight loss (A vs. B); * denotes significant main effect of protein source; *p* < 0.05 considered significant.

**Figure 5 nutrients-14-02262-f005:**
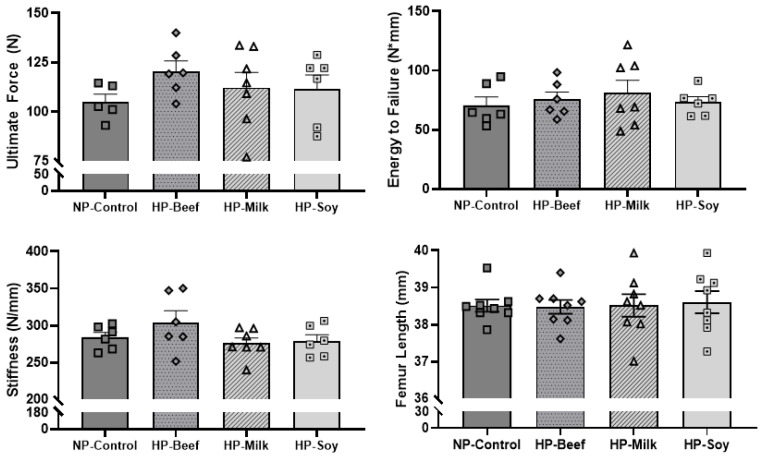
Mechanical properties of 32-week-old femurs from post-menopausal rodents following 12 weeks of energy restriction. Mean ± SEM; One-way ANOVA was performed.

**Table 1 nutrients-14-02262-t001:** Energy restricted diet formulation ^1^.

Ingredient (g)	NP-Control (g)	HP-Beef (g)	HP-Milk (g)	HP-Soy (g)
** Protein Source **				
Freeze-dried lean beef powder	0	264.8	0	0
SUPRO ^®^ 661 soy protein isolate	0	0	0	218
IdaPro milk protein isolate	0	0	230	0
Casein	140	140	140	140
DL-Methionine	0	0	0	3
L-Cystein	1.8	1.8	3	1.8
** Carbohydrates **				
Corn Starch	495.7	272	264	274.5
Maltodextrin 10	125	125	125	125
Sucrose	100	100	100	100
Cellulose	50	50	50	50
** Fat **				
Soybean Oil	40	0	37.5	40
TBHQ	0.008	0.008	0.008	0.008
** Micronutrients **				
Mineral Mix S10022M	59	59	59	59
Vitamin Mix V10037	17	17	17	17
Choline Bitartrate	4.2	4.2	4.2	4.2
** Total **	1032.7	1033.8	1029.7	1032.5

^1^ AIN-93M modified diet formulations. TBHQ: tert-Butylhydroquinone.

**Table 2 nutrients-14-02262-t002:** Macro- and micronutrient composition of energy-restricted diets ^1^.

	NP-Control	HP-Beef	HP-Milk	HP-Soy
Energy, kcal/g	3.7	3.6	3.6	3.6
Protein, %kcal	15	35	35	35
Carbohydrate, %Kcal	75	55	56	56
Fat, %kcal	10	11	10	10
Protein, g/kg	124	323	323	323
Carbohydrate, g/kg	731	516	515	516
Fat, g/kg	40	45	40	40
Fiber, g/kg				
Calcium, g/kg	8.2	8.2	12.9	9.1
Phosphate, g/kg	4.3	4.3	6.7	6.0
Potassium, g/kg	5.9	5.9	6.7	6.3
Sulfur, g/kg	1.5	1.5	1.5	1.5
Magnesium, g/kg	0.8	0.8	0.8	0.8
Sodium, g/kg	1.6	1.6	1.9	1.6
Chloride, g/kg	2.7	2.7	2.8	2.7
Vitamin A, IU/kg	6610	6610	6620	6612
Vitamin D3, IU/kg	1646	1646	1693	1646
Vitamin E, IU/kg	123	123	123	123
Total isoflavones, mg/kg	n.d.	n.d.	n.a	371

^1^ n.a.: Not available; n.d.: Not detectable.

**Table 3 nutrients-14-02262-t003:** Body Composition changes following 12 weeks of energy restriction in obese ovariectomized Sprague Dawley females ^1^.

	NP-Control	HP-Beef	HP-Milk	HP-Soy	Significance ^2^	
					Weight Loss	Diet
Body Weight, g						
Baseline	520 ± 28	505 ± 38	514 ± 23	518 ± 33		
Post	369 ± 27	385 ± 34	372.5 ± 18	397 ± 28		
Change	−124 ± 6	−103 ± 6	−112 ± 5	−98 ± 5	**<0.0001**	**0.003**
Fat Mass, mm^2^						
Baseline	2463 ± 198	2330 ± 255	2503 ± 161	2496 ± 262		
Post	1040 ± 210	1038 ± 268	1049 ± 164	1291 ± 269		
Change	−1415 ± 74	−1286 ± 62	−1447 ± 54	−1179 ± 47	**<0.0001**	**0.042**
Muscle Mass, mm^2^						
Baseline	392 ± 7	359 ± 14	366 ± 8	360 ± 11		
Post	357 ± 15	346 ± 20	355 ± 12	350 ± 17		
Change	−34 ± 11	−13 ± 13	−11 ± 12	−5 ± 10	**0.012**	0.501
Muscle Density, g/cm^2^						
Baseline	80.6 ± 0.5	80.4 ± 0.5	80.4 ± 0.4	80.6 ± 0.7		
Post	83.0 ± 0.6	83.0 ± 0.5	83.3 ± 0.5	82.2 ± 0.7		
Change	2.5 ± 0.5	2.8 ± 0.3	3.0 ± 0.4	1.8 ± 0.5	**<0.0001**	0.235
IMAT, mm^2^						
Baseline	109 ± 5	105 ± 8	104 ± 5	105 ± 5		
Post	93 ± 8	90 ± 8	95 ± 6	96 ± 6		
Change	−16 ± 4	−15 ± 2	−9 ± 3	−9 ± 4	**<0.0001**	0.415

^1^ Mean ± SEM. Change (Post-Baseline); IMAT: Intramuscular adipose tissue. ^2^ Repeated measures ANOVA: Main effect of Weight Loss and Diet. Post-hoc analysis utilized Tukey’s HSD to reveal Diet Effect. Significance denoted by bolding (*p* < 0.05).

**Table 4 nutrients-14-02262-t004:** Bone mineral density and bone mineral content changes following 12 weeks of energy restriction in obese ovariectomized Sprague Dawley females ^1^.

	NP-Control	HP-Beef	HP-Milk	HP-Soy	Significance ^2^	
					Weight Loss	Diet
Spine a BMD, g/cm^2^						
Baseline	0.1617 ± 0.0056	0.1639 ± 0.0065	0.1603 ± 0.0049	0.1567 ± 0.0041		
Post	0.1464 ± 0.0072	0.1393 ± 0.0059	0.1441 ± 0.0065	0.1372 ± 0.0062		
Change	−0.0153 ± 0.0042	−0.0246 ± 0.0061	−0.0162 ± 0.0056	−0.0195 ± 0.0068	**<0.0001**	0.625
Femur aBMD, g/cm^2^						
Baseline	0.2207 ± 0.0075	0.2362 ± 0.0056	0.2216 ± 0.0062	0.2278 ± 0.0079		
Post	0.2121 ± 0.0043	0.2097 ± 0.0046	0.2046 ± 0.0059	0.2168 ± 0.0028		
Change	−0.0109 ± 0.0066	−0.0264 ± 0.0027	−0.0171 ± 0.054	−0.0120 ± 0.0084	**<0.0001**	0.372
Spine aBMC, g						
Baseline	0.4579 ± 0.0236	0.4523 ± 0.0204	0.4650 ± 0.0136	0.4478 ± 0.0142		
Post	0.4076 ± 0.0320	0.3687 ± 0.0271	0.3760 ± 0.0218	0.3788 ± 0.0212		
Change	−0.0503 ± 0.0205	−0.0837 ± 0.0234	−0.0891 ± 0.0167	−0.0690 ± 0.0217	**<0.0001**	0.657
Femur aBMC, g						
Baseline	0.45988 ± 0.0173	0.48905 ± 0.0166	0.4728 ± 0.0113	0.4709 ± 0.0231		
Post	0.4734 ± 0.0095	0.4635 ± 0.0164	0.4663 ± 0.0077	0.4765 ± 0.0044		
Change	0.0042 ± 0.0125	−0.0256 ± 0.0066	−0.0066 ± 0.0101	0.0052 ± 0.0194	0.504	0.404

^1^ Mean ± SEM, unadjusted for baseline body weight. Change (Post-Baseline). ^2^ Repeated measures ANOVA: Main effect of Weight Loss and Diet. Post-hoc analysis utilized Tukey’s HSD to reveal Diet Effect. Significance denoted by bolding (*p* < 0.05).

**Table 5 nutrients-14-02262-t005:** Main effect of diet on bone microarchitecture following 12 weeks of energy restriction ^1^.

	NP-Control	HP-Beef	HP-Milk	HP-Soy	Diet Effect ^2^
Distal Metaphysis					
VcaBMD (mg HA/ccm)	821 ± 6	818 ± 9	818 ± 5	813 ± 6	0.869
Tb BMC (mg HA × 10^2^)	0.82 ± 0.05	0.62 ± 0.06	0.88 ± 0.10	0.96 ± 0.05	**0.012**
BV/TV × 10^2^	9.80 ± 1.22	7.09 ± 1.11	10.00 ± 1.22	11.56 ± 0.06	**0.044**
Tb.N (mm^−1^)	1.05 ± 0.10	1.07 ± 0.20	1.14 ± 0.10	1.29 ± 0.14	0.639
Tb.Th (mm)	0.110 ± 0.004	0.105 ± 0.006	0.106 ± 0.002	0.110 ± 0.004	0.772
Tb.Sp (mm)	0.86 ± 0.02	1.19 ± 0.20	0.87 ± 0.05	0.83 ± 0.08	0.128
SMI	2.16 ± 0.14	2.42 ± 0.09	2.15 ± 0.10	1.96 ± 0.05	**0.042**
Conn.D (mm^3^)^−1^	14.80 ± 2.24	10.98 ± 2.95	14.51 ± 2.76	16.20 ± 1.43	0.482
Midshaft Diaphysis					
VctBMD (mg HA/ccm)	1301 ± 4	1298 ± 2	1302 ± 4	1290 ± 3	0.061
Tt.Ar (mm^2^)	10.71 ± 0.30	10.92 ± 0.35	10.58 ± 0.29	11.25 ± 0.32	0.479
Ct.Ar (mm^2^)	6.41 ± 0.22	6.57 ± 0.09	6.49 ± 0.15	6.50 ± 0.14	0.931
Ct.Ar/Tt.Ar	0.60 ± 0.013	0.606 ± 0.020	0.615 ± 0.015	0.581 ± 0.021	0.590
Ma.Ar (mm^2^)	4.29 ± 0.20	4.35 ± 0.38	4.10 ± 0.25	4.75 ± 0.34	0.457
Ct.Th (mm)	0.599 ± 0.015	0.640 ± 0.017	0.637 ± 0.015	0.599 ± 0.030	0.323
pMOI (mm^4^)	16.83 ± 1.18	16.82 ± 0.80	17.24 ± 0.61	17.74 ± 0.69	0.850
Vertebral Body					
VcaBMD (mg HA/ccm)	911 ± 10	912 ± 12	916 ± 10	917 ± 9	0.961
Tb BMC (mg HA × 10^2^)	0.89 ± 0.06	0.79 ±0.11	0.92 ± 0.04	0.91 ± 0.08	0.590
BV/TV × 10^2^	30.33 ± 1.93	28.13 ± 2.99	32.18 ± 1.53	31.05 ± 2.33	0.639
Tb.N (mm^−1^)	2.93 ± 0.09	2.77 ± 0.27	2.94 ± 0.14	2.95 ± 0.21	0.906
Tb.Th (mm)	0.102 ± 0.003	0.097 ± 0.004	0.104 ± 0.003	0.102 ± 0.003	0.463
Tb.Sp (mm)	0.338 ± 0.022	0.381 ± 0.048	0.332 ± 0.021	0.337 ± 0.033	0.696
SMI	0.31 ± 0.09	0.67 ± 0.11	0.30 ± 0.10	0.48 ± 0.14	0.111
Conn.D (mm^3^)^−1^	40.52 ± 3.95	38.73 ± 4.91	36.13 ± 2.99	35.63 ± 3.09	0.782

^1^ Mean ± SEM. Conn.D, connectivity density; Ct.Ar/Tt.Ar, cortical area fraction; Ct.Ar, cortical bone area; Ct.Th, cortical thickness; pMOI, polar moment of inertia; Ma.Ar, marrow area; SMI, structural model index; Tt.Ar, total area; Tb BMC, trabecular BMC; Tb.N, trabecular bone number; Tb.Sp, trabecular bone separation; Tb.Th, trabecular bone thickness; BV/TV; trabecular bone volume fraction; VcaBMD, volumetric cancellous BMD; VctBMD, volumetric cortical BMD. ^2^ One-way ANOVA: Main effect of Diet. Post-hoc analysis utilized Tukey’s HSD; Significance denoted by bolding (*p* < 0.05).

**Table 6 nutrients-14-02262-t006:** Bone histomorphometry measurements following 12 weeks of energy restriction ^1^.

	Baseline	NP-Control	HP-Beef	HP-Milk	HP-Soy	Significance ^2^	
						Weight Loss	Diet
Dynamic Histomorphometry							
Periosteal MS/BS (%)	52.3 ± 6.7	40.1 ± 8.0	53.9 ± 4.2	59.5 ± 7.5	46.9 ± 7.7	0.381	0.279
PerioSteal MAR (μm/day)	0.396 ± 0.012	0.423 ± 0.015	0.385 ± 0.018	0.372 ± 0.040	0.419 ± 0.047	0.732	0.687
Periosteal BFR/BS (μm^3^/μm^2^/day)	75.4 ± 9.1	101.6 ± 8.6	86.5 ± 10.5	73.2 ± 12.0	87.8 ± 22.0	0.691	0.675
Endocortical MS/BS (%)	13.2 ± 2.2	29.4 ± 3.9	29.6 ± 5.6	19.4 ± 4.2	15.5 ± 1.9	**0.019**	**0.047**
Endocortical MAR (μm/day)	0.332 ± 0.061	0.403 ± 0.043	0.414 ± 0.055	0.397 ± 0.051	0.387 ± 0.048	0.901	0.924
Endocortical BFR/BS (μm^3^/μm^2^/day)	14.5 ± 5.8	36.8 ± 14.0	52.5 ± 11.9	30.4 ± 8.2	21.3 ± 3.1	**0.016**	0.226
Static Histomorphometry							
BS (cm)	4361 ± 381	2423 ± 362	2362 ± 482	2898 ± 359	3305 ± 97	**0.018**	0.370
O.S (mm)	1480 ± 266	1173 ± 285	1056 ± 205	1206 ± 181	1657 ± 162	0.308	0.237
O.Th (mm^2^)	2.36 ± 0.04	2.17 ± 0.07	2.21 ± 0.07	2.33 ± 0.09	2.24 ± 0.08	0.347	0.474
O.S/BS (%)	2.8 ± 0.1	4.6 ± 0.7	4.8 ± 0.6	4.6 ± 1.0	5.0 ± 3.7	0.431	0.237
E.Pm (mm)	6.6 ± 2.2	2.7 ± 0.7	2.6 ± 0.7	3.0 ± 0.5	3.6 ± 0.4	**0.041**	0.601
ES/BS (%)	13.0 ± 1.6	12.8 ± 1.9	14.7 ± 1.6	12.1 ± 1.7	11.7 ± 1.4	0.509	0.647
N.Oc	182 ± 65	99 ± 10	59 ± 17	64 ± 16	87 ± 4	**0.017**	0.202
Oc.S/BS (%)	4.8 ± 0.5	5.1 ± 0.5	5.5 ± 0.8	3.5 ± 0.4	4.9 ± 0.7	0.213	0.152
N.Oc/E.Pm (Ratio)	27.3 ± 0.6	31.4 ± 6.1	23.7 ± 1.2	21.2 ± 3.4	27.2 ± 2.2	0.405	0.381

^1^ Mean ± SEM. BFR, bone formation rate; BS, bone surface; E.Pm, eroded perimeter; ES, eroded surface; MAR, mineral apposition rate; MS, mineralizing surface; N.Oc, osteoclast number; Oc.S, osteoclast surface; O.S, osteoid surface; O.Th, osteoid thickness. ^2^ Two-way ANOVA: Main effect of Weight Loss and Diet. Significance denoted by bolding (*p* < 0.05); Post-hoc analysis utilized Tukey’s HSD.

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
