# Peer review of "Effects of Dietary Protein Source and Quantity on Bone Morphology and Body Composition Following a High-Protein Weight-Loss Diet in a Rat Model for Postmenopausal Obesity"

_nutrients, 2022, doi:10.3390/nu14112262_

Round 1

Reviewer 1 Report

The paper is very interesting and the conclusion are supported by the data. However the data presentation could be improved to be clearer for the reader. Many tables with p values only between the 4/5 groups, bold when it is significantly different or stars, but also stars to denote a trend: this is confusing. To me, more graph would be easier to show what important differences you found between the different groups, with * between the groups. If you look in the same time for the weight loss effect and the diet effect, should you rather use a 2-way ANOVA?

also, many abbreviations that are not so easy to understand at first: NP, HP, WL, ER ...

The study is very interesting, but you do not really conclude on the usefulness of HP vs NP. It seems to me there is not a big clear effect of HP vs NP diet.

More specifically:

 - page 3 line 99: 45% high fat and 22% high sugar diet, how much increase in % does it make compared to normal diet?

  • was there an effect on the animals behavior to stop HFHS diet? Was is addictive to them before you reduced this diet?
  • page 4 diet formulation paragraph: was the powder you added designed as pellets?
  • page 4 line 133: could you add the carbohydatres substituion in % to be able to compare with the HFHS diet please?
  • page 4 line 133: Campbell study showed decreased BMD while using animal based proteins source. Why would you xant to reproduce it? Please explain better
  • page 5: did you check the food eaten by rats every day? where they in individual cages?
  • page 6: I am very used to µCT data but line 178 you speak of B. What is it please? You should describe it since it is not a common abbreviation. What does it represent? Bone surface? volume?
  • page 6 line 176: muscle density is a very interesting parameter. You do not discuss the change much in your discussion.
  • page 6 line 197: what s aerial BMD? how is it measured please?
  • page 7 line 211: 25 slices total analyzed or 25 proximally and 25 distally which make a total of 50?
  • page 7 line 215: threshold defined similar to previous studies: the thresholds should be written, since other study probably have other devices than you.
  • page 7: histology paragraph: how were the sections cut? with which device? von Kossa / Mac Neal's modified protocol to reveal what? data shown where in the graph/ tables?? same for the dynamic histomorphometry paragraph below. slices cut with what? which thickness?
  • page 8: analysis: why not 2 way ANOVA used ?
  • why SE and not SEM?
  • page 8 line 280: why WL increased muscle density??
  • table 3: not clear p value for ANOCA test. Then which group is different from which? We can not know it by looking at the graph only
  • page 10: mispelling in the title "femur" line 293
  • page 10: LS.aBMD, F.aBMD, not easy to understand. Could you soimply write lumbar spine and femur BMD?
  • Page 10: why show the data in the table + a graph? To me the graph is clearer
  • page 10: not need to describe the stat done in the figure legend. It has been done in the paragraph statistics already
  • table 4: legend F= femur, but in the table RF. Means right femur? please simplify
  • page 11: bone loss seen could be an effect of aging? at what age the rat start to loose trabecular bone with aging?
  • page 11: same comment; Why graph + table? it makes a lot of data to read. A lot clearer with the figure 4 to see the differences between groups.
  • A different from B, * for difference between diets. Not easy to understand at first. The results should not be described in the fugure legend.
  • table 5 page 12: significance in bold this time and for trends. You can not use bold, or A vs B and stars for significance on a graph and for trends on another graph. This is very confusing. Please keep the same for all the paper.   stars means significantly different in most papers.
  • what difference between table 5 and table 6? Legend of table 5 is post hoc test reveal dief effect, but the diet effects are shown in table 6 ... this is confusing. I do not understand with the graph easily which group is different from which group.
  • page 12 SMI lowered line 331: is it good or bad ?? rod or plate like trabeculae are correlated to bone fracture?
  • page 12 line 332: trend for volumetric cortical BMD observed for HP-Milk vs HP-soy. Not for HP-beef? the value is the same as for HP-milk (818)
  • figure 5 page 14: would have been interesting to have a baseline control, to compare the loss of resistance/mechanical strength with weight loss and the different diets
  • page 14 line 348; why only trends with post hoc test for encortical MS/BS while the p value for ANOVA is p=0.047?
  • page 14 line 360: you speak of diet effect (beef) on the oC number while this time the p value for diet effect is not sigificant (p=0.202). Why is that you have a diest effect than?
  • page 14 line 363. No effect in osteoclast activity relative to bone surface. Why is that? time of the study not long enough? Effect on cell activity recent and not yet shown at the tissue level?
  • table 7 legend/ please remove star denotes a trend since 1/it is confusing and 2/ there is no trend in this table
  • page 15 line 380 conclusion: I would not speal of unloading, but of decreased load on bone due to decreased weight (like 20% loss).
  • page 15 line 390: which parameters do you refer to when you speak of higher bone turnover rates? please explain since you have so many data in this paper. The HP-beef diet exacerbated bone loss, but no BMC, BMD change, N.Oc lower, MS/BS higher, BFR/BS higher .... be careful with your conclusion
  • page 16 line 417: attenuated decreased quality with HP-soy diet: please explain which data  you use to say that. The N.Oc is higher in soy than beef...
  • page 16 line 445: HP-milk showed a trend for greater volumetric BMD...shown where? lower than HP-soy in table 5 and no diet effect on this paramer in table 6.
  • page 17 line 460: you can cite your table 3 where you saw a difference in fat mass loss among the diets, HP-milk leading to higher fat mass loss.
  • I do agree that power calculations do give weird results whe you analyse many parameters, like you should use 80 animals per group .... which we cannot do for ethical and financial reasons. But not folowwing the power calculations is not because your research quetsion is novel I believe. I had the same problems many years ago!
  • conclusion: you do not discuss if there is a real interest of HP vs NP diet ... HP should be very usefull to be used since it is probably more expensive for people
  • question: loss of BMC and BMD in obese women with weight loss, but are their value after the weight loss comparable to normal weight women?

Author Response

Please see the attached word document. Thank you for your insights. 

Reviewer 2 Report

Comments to the Authors of manuscript number: nutrients-1730948 entitled “Effects of dietary protein source and quantity on bone morphology and body composition following a high-protein weight-loss diet in a rat model for postmenopausal obesity”.

The manuscript presents the study performed on obese ovariectomized rats subjected to different types of diet to induce the weight loss. As a WL diet were used High protein diets with different sours of protein, milk, soy or beef. It is very interesting, well performed and described study.

  1. L 21 although in L 17 is used “ER” but in the form of “HP-ER”. It can be assumed that ER means energy restricted, but readers have to be sure
  2. L 22 – NP-?
  3. L 22 – there is not clear how many diets are presented
  4. L 99 – to what type of diet it relates?
  5. part of 2.8 were these bones the same which were used to CT analysis and histology?
  6. L 230 -were decalcified? In further part it is explained by the stained protocol, but is should be written clearly
  7. mechanical testing related to midshaft, CT to distal or proximal part?
  8. If Authors wrote that main bone loss relates to the hip part of femur, why histology is performed to distal part?
  9. I understand, that this study is finished, but why there were not detected main markers of bone turn over or hormones related to obesity like leptin?

Author Response

Comments to the Authors of manuscript number: nutrients-1730948 entitled “Effects of dietary protein source and quantity on bone morphology and body composition following a high-protein weight-loss diet in a rat model for postmenopausal obesity”.

The manuscript presents the study performed on obese ovariectomized rats subjected to different types of diet to induce the weight loss. As a WL diet were used High protein diets with different sours of protein, milk, soy or beef. It is very interesting, well performed and described study.

L 21 although in L 17 is used “ER” but in the form of “HP-ER”. It can be assumed that ER means energy restricted, but readers have to be sure

  • ER has been changed to “energy restriction” for clarity

L 22 – NP-?

  • Line 23: NP = Normal protein

L 22 – there is not clear how many diets are presented

  • There are 4 energy restricted diets as described in Line 21

L 99 – to what type of diet it relates?

  • The AIN-93M diet is a standardized chow diet given to all adult rodents in captivity (Reeves PG et al, 1993 J Nutrition)

Part of 2.8 were these bones the same which were used to CT analysis and histology?

  • The femurs used for mechanical testing were separate from those used for CT analyses and histology. The right femur was used for CT and histology, while the left femur was used for mechanical testing (Line 170-173)

L 230 -were decalcified? In further part it is explained by the stained protocol, but is should be written clearly

  • Both the von Kossa / MacNeal’s (VKM) Tetrachrome staining protocol (reference 80) and the tartrate-acid resistant acid phosphatase (TRAP) staining protocol (reference 81) are well-established and standardized procedures in bone histology. As such, a more detailed description of the protocol is not warranted in the manuscript.

Mechanical testing related to midshaft, CT to distal or proximal part?

  • Both the “Distal metaphysis and midshaft diaphysis of right femora” (Line 216-217) were analyzed with uCT to assess changes in cancellous and cortical bone structure.
  • Testing the mechanical strength of femora by three-point bending relates to the midshaft femur. However, this is now further clarified in the methods section (Line 239)

If Authors wrote that main bone loss relates to the hip part of femur, why histology is performed to distal part?

  • Thank you for your question. Though uCT measurements of the proximal femur can be taken, common histological procedures are not usually done at the proximal femoral site due to its unique shape and poor sectioning. However, the distal femur can have both its microarchitecture and cellular activity assessed through uCT and histological staining.

I understand, that this study is finished, but why there were not detected main markers of bone turn over or hormones related to obesity like leptin?

  • Thank you for your question. This study was funded through a small pilot grant and as a result did not have the resources to assess circulating bone turnover markers nor hormonal levels.
  • This now is included as a limitation to the study (Line 521-524)

Reviewer 3 Report

I have reviewed the manuscript entitled “Effects of dietary protein source and quantity on bone morphology and body composition following a high-protein weight-loss diet in a rat model for postmenopausal obesity” submitted for a publication in Nutrients as an original article. However, Authors didn't draft the paper well .

L110 A major concern – As the number of OVX animals at baseline was n=4 and they were randomized to one of four energy-restricted diets it means that for each group the baseline measurements were done only for one animal. How the statistical analysis, can be performed when having only one subject per group ??

Figure 5 A major concern - what was the number of examined bones ? 5 for control, 6 for beef and soy and 8 for milk? why? also why there are only 5 symbols for soy for strain, (Young’s) modulus and toughness while for force, energy and stiffness 6 points are marked?

Figure 5 A major concern - the results for strain are completely messed up - strain is equal to the fraction of bone deformation or the percentage of bone deformation. It is impossible to get the values of the order of tens of thousands.

L49 I have carefully read Sherman's work [16] and I wasn't able to find any mentions of bone-ash hypothesis. Please use different, more appropriate and novel references like meta-analysis  doi.org/10.1359/jbmr.090515 or review doi.org/10.1007/s00774-014-0571-0

L107 Please add the approval number

Table1 footnote: There is no “1” superscript in table 1

Figure 1 – please mark (list) which measurements were performed in vitro (post mortem) and which in vivo (under anesthesia) . Please add the information of n number for each measurement per group

L140 mTorr

L165 and L193 These lines show two different anesthesia schemes with the same reference. Please verify.

L195 and L196 areal ?

L205 kVp

L213-214 If you describe all threshold values in such detailed manner in 2.6, please add the appropriate threshold values also there, not only refer to other study

L225 ref [75] does not describe bone mechanical testing

L223 What about strain, modulus (Young’s modulus?) and toughness? How were they calculated ?

L226 compression is not the same stress as bending. Please rephase

L240 TRAP

L244 Again, please list the used nomenclature

L253-255 Any references available for these equations? I suppose that they describe rat-specific deposition and formation rates, which probably are also dependent on epiphysial location.

Figure 3 repeats the data from table 4 and therefore should be removed

L312 “Energy restriction decreased measurements of bone microarchitecture” – please rephase

Tables footnotes- A lot of typo errors. For example: Ma.r vs Ma.Ar, TB BMC vs TRAB BMS. Also the order of abbreviations is completely incomprehensible

Table 5 – Please use the same group naming as in the manuscript.

Table 7 - it would be beneficial to include as supplementary data the representative images, of dynamics histochemistry (not for all groups, just to show the measurement methodology)

Table 7 - remove from  table footnote the information about the trend - no trend values can be observed  in table 7

L424-425 Was it measured in this study ?

L466 – Other limitations- the lack of the analysis of bone turnover markers

References – please verify the citations as ref. 73 and 74 duplicate

Trend was not defined in M&M section.

Throughout the manuscript, please add the space between the numerical value and corresponding unit of physical quantity. Also please unify the nomenclature. For example BV/TV: trabecular bone volume fraction vs relative bone volume; tb.th trabecular bone thickness vs trabecular thickness

Layout of some references is not in accordance with journal recommendations / specifications

Author Response

I have reviewed the manuscript entitled “Effects of dietary protein source and quantity on bone morphology and body composition following a high-protein weight-loss diet in a rat model for postmenopausal obesity” submitted for a publication in Nutrients as an original article. However, Authors didn't draft the paper well 

L110 A major concern – As the number of OVX animals at baseline was n=4 and they were randomized to one of four energy-restricted diets it means that for each group the baseline measurements were done only for one animal. How the statistical analysis, can be performed when having only one subject per group ??

  • Thank you for your question. The animals sacrificed at baseline (OVX=4, Sham=4) were used to confirm ovariectomy (OVX vs. Sham) via estrogen-deficiency osteoporosis, and to establish baseline µCT and histological measurements before the weight loss intervention (baseline OVX=4). The remaining 32 rats were then randomized into one of four groups (n=8 per group) and put on an energy restrictive diet. Baseline sacrifices occurred prior to randomization, and therefore serve as a baseline comparison for µCT and histology measurements.

Figure 5 A major concern - what was the number of examined bones? 5 for control, 6 for beef and soy and 8 for milk? why? also why there are only 5 symbols for soy for strain, (Young’s) modulus and toughness while for force, energy and stiffness 6 points are marked?

  • Thank you for bringing up this concern. The difference in sample sizes between multiple strength outcomes is due to the presences of outliers in the original dataset that were removed according to the outlier labeling rule. This was not originally stated in the statistical analysis section, but is now included (Line 281-283)

Figure 5 A major concern - the results for strain are completely messed up - strain is equal to the fraction of bone deformation or the percentage of bone deformation. It is impossible to get the values of the order of tens of thousands.

  • Thank you very much for bringing this to our attention. The y-axis was mislabeled in the original submission and should have been labeled microstrain (µε), which would explain the values ranging from 10 to 35 thousand. Strain values (ε or ΔL/L) for rodent femurs typically lie between 0.01 and 0.03, which is inline with our original submission. However, after careful consideration and for the sake of clarity, we’ve decided to remove Total Strain, Modulus, and Toughness graphs from Figure 5 as these outcomes provide little addition information beyond what’s included with Ultimate Force, Energy to Failure, and Stiffness. Additionally, femur length was added to show it did not influence mechanical strength outcomes.

L49 I have carefully read Sherman's work [16] and I wasn't able to find any mentions of bone-ash hypothesis. Please use different, more appropriate and novel references like meta-analysis  doi.org/10.1359/jbmr.090515 or review doi.org/10.1007/s00774-014-0571-0

  • The intent was to include the original publication sighting urinary calcium excretion following the consumption of excess dietary protein. However, the reviewer makes an excellent point, and the citation has been replaced with the 2009 meta-analysis.

L107 Please add the approval number

  • The approval number and approval date have now been added (line 113-114).

Table1 footnote: There is no “1” superscript in table 1

  • Added

Figure 1 – please mark (list) which measurements were performed in vitro (post mortem) and which in vivo (under anesthesia). Please add the information of n number for each measurement per group

  • As noted in both Body Composition and DXA method sections, rodents were anesthetized during acquisition of measurements in vivo. Femoral strength and histomorphometry can only be assessed ex vivo (outside of the living body). However, micro-computed tomography can be assessed by in vivo and ex vivo, therefore this clarification is now added to Figure 1.
  • The sample size for energy restricted diet is now included in Figure 1 as well.

L225 ref [75] does not describe bone mechanical testing

  • Cui et al 2011 (reference 76) describes bone mechanical testing in the “Assessment of bone properties” method section as well as additional details in supplementary materials.

Figure 3 repeats the data from table 4 and therefore should be removed

  • Figure 3 illustrates change values between the 4 bone quantity outcomes, while Table 4 provides baseline and post-intervention means & SEM, along with p-values of two-way ANOVAs.

L466 – Other limitations- the lack of the analysis of bone turnover markers

  • This study was funded through a small pilot grant and as a result did not have the resources to assess circulating bone turnover markers nor hormonal levels. This limitation is now added to the discussion section (Line 521-524)

Throughout the manuscript, please add the space between the numerical value and corresponding unit of physical quantity. Also please unify the nomenclature. For example BV/TV: trabecular bone volume fraction vs relative bone volume; tb.th trabecular bone thickness vs trabecular thickness

Layout of some references is not in accordance with journal recommendations / specifications

Round 2

Reviewer 3 Report

The Authors have answered all my main issues which were raised during first review.